# Investigation of Multiple Shape Memory Behaviors, Thermal and Physical Properties of Benzoxazine Blended with Diamino Polysiloxane

**DOI:** 10.3390/polym15183814

**Published:** 2023-09-18

**Authors:** Sunan Tiptipakorn, Chanikan Angkanawarangkana, Sarawut Rimdusit, Kasinee Hemvichian, Pattra Lertsarawut

**Affiliations:** 1Department of Physical and Material Sciences, Faculty of Liberal Arts and Science, Kasetsart University, Nakhon Pathom 73140, Thailand; chnkfern@gmail.com; 2Center of Excellence in Polymeric Materials for Medical Practice Devices, Department of Chemical Engineering, Faculty of Engineering, Chulalongkorn University, Bangkok 10330, Thailand; sarawut.r@chula.ac.th; 3Thailand Institute of Nuclear Technology (Public Organization), Ongkarak District, Nakhon Nayok 26120, Thailand; kasinee@tint.or.th (K.H.); pattra@tint.or.th (P.L.)

**Keywords:** multiple shape memory systems, benzoxazine, polysiloxane

## Abstract

In this research, benzoxazine (BA-a) and diamino polysiloxane (PSX750) blends were prepared at 0–50 wt% of BA-a. The interactions between two polymeric components were investigated via a Fourier Transform Infrared Spectrometer (FT-IR). The thermal properties of the blends were also determined with Dynamic Mechanical Analyzer (DMA) and Thermogravimetric Analyzer (TGA). The mechanical properties and shape memory behaviors of the blends were also investigated. The FTIR spectra exhibited the shift of the peak from 1672 to the range of 1634–1637 cm^−1^, which could be identified as hydrogen bonds between two polymeric domains at the contents from 30 to 50 wt%. The DMA thermograms revealed two glass transition temperatures, which could indicate a partially miscible system. The char yield values were increased, while the decomposition temperatures were decreased with an increasing benzoxazine content. Interestingly, the blends at the contents of 10 and 20 wt% presented dual-shape memory behaviors, whereas triple- or multiple-shape memory behaviors were observed with benzoxazine contents of 30 to 50 wt%. For the high-temperature recovery state, a shape memory ratio of 97.5% with a recovery time of 65 s and a shape fixity ratio of 66.7% was recorded at the content of 50 wt%. For the low-temperature recovery state, a shape recovery ratio of 98.9% was observed at the same content. Moreover, the values of the recovery ratio for four shape-recovery cycles revealed multiple shape memory behaviors with high recovery ratios in the range of 95–98%.

## 1. Introduction

Shape memory polymers (SMPs) are known as a class of smart polymeric materials, which are able to be deformed and fixed in a temporary shape when an external trigger is applied. They can recover their original shape when the proper stimulus is used. SMPs can be generally stimulated by heat, pH (the level of alkalinity or acidity), moisture, and magnetic or electric fields [1,2]. In recent years, SMPs have attracted great attention because of their wide range of uses including self-healing materials, smart fabric and fibers, biomedical devices, and self-deployable structures for aerospace applications [3,4,5]. For the last few decades, thermosetting shape memory polymers based on polybenzoxazine have drawn great attention because polybenzoxazine possesses interesting properties with a chemical crosslinking structure, which could be determined as a permanent shape in SMPs [6,7,8].

Polybenzoxazine, a novel kind of phenolic thermoset resin, possesses many benefits such as the ability to be cured without the need for a catalyst via ring-opening polymerization. Moreover, there are no byproducts, which could generate voids in the products upon curing. After processing, the polybenzoxazine products have near zero volumetric shrinkage with excellent thermal stability, high mechanical properties, and low water absorption [9,10,11,12]. Nowadays, bisphenol A-aniline benzoxazine (BA-a) resin is commercially available. However, the use of this thermosetting was found to be limited in industrial applications due to its brittleness. In general, the toughness of polybenzoxazine could be improved by two main approaches, i.e., the first one is the molecular design of benzoxazine resins [13] and the second one is blending or alloying benzoxazine with other polymeric components. The latter approach is a comparatively simple and versatile method to enhance the mechanical and thermal characteristics of polybenzoxazine. Recently, polybenzoxazine (PBA-a) was developed by blending and alloying it with other polymers such as polyurethane, epoxy, polyimide, dianhydride, etc., in order to decrease the rigidity of the thermosetting polymer [14]. Shape memory polybenzoxazine copolymerized with apolyurethane prepolymer and epoxy has been developed in recent years. In these blending systems, polybenzoxazine is the hard segment-rich domain due to aromatic rings in its chemical structure, while polyurethane and epoxy are the soft segment-rich domain [15,16,17].

Polysiloxanes or silicone polymers are well-known as versatile industrial materials utilized for sealing, coating, paints, electronics, cosmetics, and construction applications due to their beneficial aspects such as great thermal stability, good insulation properties, and high flexibility [18,19,20,21,22]. Applications of polysiloxane in shape memory systems have been conducted for decades. In 2007, Keller et al. studied the modified polydimethylsiloxane (PDMS) elastomer as shape memory materials for self-healing applications with approximately 70–100% recovery. However, the limitation of the system is that only one time of the healing for a given location was reported [20]. Later, Zhao et al. studied the system of the poly(siloxane-urethane) elastomer with high tensile strength and dual-shape memory behaviors [21]. Recently, shape memory polybenzoxazines based on a siloxane-containing diphenol were prepared by incorporating Si-O-Si linkage into the structure of benzoxazine; therefore, the flexible portions were introduced in the polybenzoxazine. One-way dual-shape memory behaviors were reported [22]. In this study, diamino polysiloxane has been proposed to blend with benzoxazine in order to increase the usage range of thermosetting materials for shape memory behaviors without structure modification and with a relatively simple method. The multiple-shape memory behaviors and thermal and physical properties of the blending systems were also studied.

## 2. Experimental

### 2.1. Materials and Methods

The materials in this study are benzoxazine resin and diamino polysiloxane. Benzoxazine resin is based on bisphenol-A, aniline, and formaldehyde. The bisphenol-A (polycarbonate grade) was provided by Thai Polycarbonate Co., Ltd. (TPCC, Rayong, Thailand). Para-formaldehyde (AR grade) was bought from Merck Co., Ltd. (Darmstadt, Germany) and Aniline (Reagent Plus Grade) as purchased from Sigma-Aldrich (Singapore). Bis-(3-aminoproyle) poly(dimethyl siloxane) (PSX750) were obtained from Dow Corning Toray Silicone Co, Ltd., Tokyo, Japan with the molecular weight of 750 Da. Tetrahydrofuran for the analysis of EMSURE^®^ ACS, Reag. Ph Eur was obtained from Merck Co., Ltd. (Darmstadt, Germany).

### 2.2. Preparation of Benzoxazine Monomers and PSX750/BA-a Blends

Benzoxazine (BA-a) was synthesized from bisphenol-A, paraformaldehyde, and aniline at the molar ratio of 1:4:2 according to the solventless synthesis method [23]. All components were mixed and heated at 110 °C for 45 min. The obtained solid was ground and a yellow monomer powder was obtained. The solution of PSX750 was prepared by mixing 2 g of PSX750 in 10 mL of THF. PSX750 was dissolved in THF and blended at the BA-a content of 0, 10, 20, 30, 40, and 50 wt%. The blends were poured in Teflon molds and heated in the oven at 60 °C (18 h), 100 °C (1 h), 130 °C (1 h), 150 °C (1 h), 180 °C (1 h), 205 °C (2 h), and 210 °C (0.5 h). The obtained film was left at room temperature before removing it from the mold. 

### 2.3. Characterizations

The functional groups of the obtained polymeric blends were determined using a Bruker (Tensor 27 model, Mannheim, Germany) Fourier Transform Infrared (FTIR) Spectrometer with the ATR method with a scan number of 16 and a resolution of 4 cm^−1^. The dynamic mechanical analyzer (model DMA 1 STARe system; Mettler Toledo, Greifensee, Switzerland) was used to determine dynamic mechanical properties in tensile mode. The dimensions (W × L × T) of the samples were kept at 5 mm × 25 mm × 1 mm. The deformation frequency of 1 Hz was applied. The specimen was heated at the heating rate of 2 °C·min^−1^ from −100 to 300 °C under an air atmosphere. The glass transition temperature (T_g_) was reported from the peak on the tan delta curve of DMA thermograms. The thermal stability of the blends was investigated using a Thermogravimetric Analyzer (Mettler Toledo, TGA1 STARe System, Greifensee, Switzerland) under a nitrogen flow of 80 mL/min. Approximately 10 mg samples were placed on the alumina crucible pans and then heated from 25 to 800 °C at a heating rate of 20 °C·min^−1^. The values of decomposition temperature and char yield at 800 °C were reported. Mechanical properties of the samples at room temperature were tested in tension mode using a Universal Testing Machine (Lloyd-LS1, Lloyd Instruments, Cleveland, OH, USA). A load cell of 50 N was applied. The dimensions of the tensile tested sheet were 30 (L) × 1 (W) × 1 (T) mm. The gauge length was 30 mm, and the crosshead speed was kept at 10 mm/min.

For shape memory behavior tests, all samples of the blends with the dimensions of 40 (L) × 10 (W) × 1 (T) mm were prepared in a linear shape (as the original shape) at room temperature. After that, each specimen was heated at a temperature of T_gH_ + 20 °C and bent into an L-shaped form (as the first temporary shape) under the bending force. After that, the external force was removed. After cooling at T_gL_ + 20 °C, the sample was further deformed to be a twisted shape (as the second temporary shape). The external force was then removed. The sample was kept at ca. −80 °C with dry ice. In the recovery period, each sample was left at room temperature and the recovery time for low temperature recovery state was recorded. Then each sample was further heated to T_gH_ + 20 °C; the recovery time the for high-temperature recovery state was recorded. The cycle of recovery test was conducted as shown in Figure 1.

The values of shape fixity and the shape recovery ratio were calculated using the following equations: (1)$$Shape\;fixity\left(\%\right)=\frac{\theta_{1}}{90^{{}^{\circ}}}\times 100\%$$
(2)$$Shape\;recovery\;ratio=(\frac{\theta_{1}-\theta_{2}}{\theta_{1}})\times 100\%$$
where θ_1_ is the initial angle created at 20 °C above T_gH_ and θ_2_ is the final angle after shape recovery.

For the absorption test, each sample of 10 (L) × 10 (W) × 1 (T) mm was immersed in water for one day. The values of the sample weight before and after immersion were recorded. The percentage of water absorption was calculated as follows:(3)$$\%\;water\;absorption=\frac{W_{A}-W_{B}}{W_{B}}\times 100\%$$
where W_A_ and W_B_ are the weights after and before immersion, respectively.

## 3. Results and Discussion

All blending systems were observed to be homogeneous, and there was no phase separation at the benzoxazine content of 10–50 wt%. However, the blends at contents above 50wt% were found to be immiscible; the sea-island appearances were observed to indicate phase separation in the high content of benzoxazine. The shape memory behaviors were not exhibited in the sheets of pure PSX750 and pure polybenzoxazine (PBA-a). The blending systems at low benzoxazine content, i.e., 10–20 wt% revealed only dual-shape memory behaviors for the switching temperature at approximately T_gL_ (as the second recovery step in Figure 1). It was worth noting that triple-shape memory behaviors were found at the benzoxazine content of 30–50 wt% with two switching temperatures as T_gH_ and T_gL_ (as the first and second recovery steps in Figure 1, respectively).

### 3.1. FTIR Spectra of PSX750/PBA-a Blends

The molecular structure of PSX750/PBA-a blends at various benzoxazine contents was determined by using ATR FT-IR spectroscopy. The spectra of PSX750/PBA-a blends are presented in Figure 2. 

The absorption peaks of pure PSX750 and all blends were presented at 1258, 1012, and 788 cm^−1^, corresponding to Si-CH_3_ [24], Si-O-Si [25], and N-H wagging (primary amine), respectively. However, it could be noted that the signal peak at 1672 cm^−1^ (NH stretching) [26,27] of pure PSX750 shifted with the incorporation of benzoxazine at the content of 10–50 wt%; the peak shifted to the range of 1634–1637 cm^−1^ [28]. The shifting of the signal peak of NH stretching could be due to the hydrogen bonding interactions between two polymeric domains [29,30]. For pure PBA-a, there was no signal peak at 1672 cm^−1^, but the peak at 1619 cm^−1^ (C=C stretching in aromatic) was observed [31]. The results indicated that the blends were successfully prepared. The molecular structure of PSX750/PBA-a blends was proposed as shown in Figure 3. It could be seen that the structures of benzoxazine are composed of aromatic rings leading to hard segments, while the structures of PSX750 are composed of ether linkages leading to soft segments in the shape memory system. 

### 3.2. Thermal Stability of PSX750/PBA-a Blends 

Thermal stability could inform the service temperature of materials. In general, the thermal stability values could be determined from the decomposition temperature at 5% weight loss (T_d5_) and char yield at 800 °C. From Figure 4, it could be seen that the addition of benzoxazine from 0–50 wt% could lead to an increase in char yield, i.e., an increase from 0 to 14 wt%. However, the T_d5_ values were decreased with the addition of benzoxazine, i.e., decreased from 332 to 319 °C. This could be attributed to the nature of pure polybenzoxazine, i.e., pure PBA-a provided char yield value of ca. 30 wt% and T_d5_ value of approximately 316 °C. The possible reason for the increase in char yield in the case of the blends at a higher content of benzoxazine is that the greater number of aromatic rings in the chemical structure of the benzoxazine portion led to the higher value of char yield [14].

### 3.3. Glass Transition Temperature (T_g_) of PSX750/PBA-a Blends

The glass transition temperatures of the polymeric materials are crucial parameters, with an influence on the properties of the polymers. For the shape memory polymers, the T_g_ values are relevant to the switching temperature of the materials when changing the temperature. Many well-known analytical methods could be applied for the determination of T_g_ values, such as differential scanning calorimetry (DSC) and dynamic mechanical analysis (DMA). However, from the observations of the DSC thermograms, the values could not be obviously determined; therefore, the glass transition temperature values were determined from the peak of tan delta in the dynamic mechanical analysis (DMA) technique. Table 1 presents the values of the glass transition temperature of the blends at different benzoxazine contents from the tan delta peaks; the values are averaged from the results from at least two tests. From the literature, the T_g_ of pure PBA-a was reported to be approximately 174 °C [32], while that of pure polysiloxane was reported at approximately −80 °C [33]. 

It could be noted that the blends at contents of 10 and 20 wt% exhibited only one glass transition temperature (T_gL_), at −70 and −63 °C, respectively, while two glass transition temperatures of the blends were observed at T_gL_ = −46 °C and T_gH_ = 150 °C (for 30 wt%), T_gL_ = −23 °C and T_gH_ = 157 °C (for 40 wt%), and T_gL_ = −17 °C and T_gH_ = 165 °C (for 50 wt%). This phenomenon could lead to the dual-shape memory behaviors being shown for the blends at the content of 10 and 20 wt%, whereas the multiple-shape memory behaviors were observed at the contents from 30 to 50 wt%. From the DMA thermograms, the tan delta peaks of two glass transition temperatures move toward each other. That means the partial miscibility of PSX750/PBA-a blends was shown. Furthermore, the differences between the two T_g_ values were in the wide range for the content from 30 to 50 wt%. These phenomena exhibit that the glass transition temperature of the polymeric blends could be tunable with the adjustment of benzoxazine contents. It could be noted that the difference in the two glass transition temperatures (T_gL_, and T_gH_) for 30, 40, and 50 wt% are 196, 180, and 182 °C, respectively. This suggests a high difference in the temperature range. In comparison with the two types of shape memory polybenzoxazine based on a siloxane-containing diphenol synthesized from 1,1,3,3-tetramethyldisiloxane and o-allylphenol, one glass transition temperature of the two polymers was reported at 53 and 39 °C [9]. 

### 3.4. Shape Recovery Behaviors

#### 3.4.1. Shape Fixity and Recovery Time

Shape recovery parameters obtained from the shape recovery tests of the PSX750/PBA-a blends at various benzoxazine contents are summarized in Table 2, which presents the results of the type of shape memory, shape fixity at high and low recovery temperatures, the shape recovery ratio at high and low recovery temperatures, and the recovery time at low recovery temperatures for one recovery cycle. From Table 2, the high recovery temperature means a higher switching temperature or higher glass transition temperature, while the low recovery temperature means a lower switching temperature or lower glass transition temperature, which corresponds to T_gH_ and T_gL_ in Table 1, respectively. The samples of pure PSX750 and pure PBA-a could not exhibit shape memory behaviors. This could be attributed to the fact that the materials revealed shape memory behaviors when both soft segments and hard segments are present in their chemical structures [34]. The incorporation of polysiloxane could introduce the soft segment in the blends [35], while the addition of polybenzoxazine could lead to a hard segment in the blending system [36]. The purpose of soft segments is to provide a switching ability when the polymer is heated above the glass transition temperature, whereas the hard segments could render the ability to fix the shape through physical crosslinking. Interestingly, it could be noted that the blends with a content of 10 and 20 wt% could not present the shape recovery behaviors at a high recovery temperature. They could only switch shape at a low shape recovery temperature (T_gL_) in the range of −63 to −70 °C. This switching temperature corresponded to pure PSX750. The shape recovery ratio at a low recovery temperature with a content of 10 and 20 wt% was 100% with a recovery time of 60 and 77 s, respectively. For contents of 30, 40, and 50 wt%, triple-shape memory behaviors were observed. The shape recovery ratio at a high recovery temperature was reported at 99.4, 99.3, and 97.5%, respectively. The shape fixity values at a high recovery temperature increased with benzoxazine contents, i.e., 25.6, 40.0, and 66.7%, with the recovery time in the range from 57 to 65 s, respectively. At a low recovery temperature, the shape recovery values were approximately 99% with a recovery time in the range of 78 to 360 s. From Table 1 and Table 2, it could be implied that the blends with contents from 30 to 50 wt% could be applied as multiple-shape memory polymers in a wide temperature range with high shape recovery ratios at low and high switching temperatures.

#### 3.4.2. Recovery Behaviors at Various Shape-Recovery Cycles

To explore the reusability, the cycle-dependent shape recovery ratio values, which were studied at low and high shape recovery temperatures, were determined, and are presented in Table 3. It could be noted that the shape recovery ratio of the blends at a low shape recovery temperature for blends with benzoxazine contents of 30 and 40 wt% were not significantly different in the range of ca. 97–99%, even though four shape-recovery cycles were carried out. However, the blends at 50 wt% showed a slight decrease from 98.9 to 95% at a low shape-recovery temperature after four shape-recovery cycles. For a high-shape recovery temperature and the blends at 30, 40, and 50 wt%, the recovery values were reported at 99.4, 99.3, and 97.5% for the first shape-recovery cycle. The values decreased to approximately 96% after four shape-recovery cycles. These results exhibited the reusability of the PSX750/PBA-a blends for multiple-shape memory. However, for five shape-recovery cycles, the recovery ratio at a high shape recovery temperature was drastically decreased, i.e., lower than 90%. The blending materials exhibited more rigidity. In cases of such a greater number of shape-recovery cycles, i.e., ten shape-recovery cycles, the shape memory behaviors could not be exhibited. 

### 3.5. Mechanical Properties

In order to assess the strength and modulus of multiple-shape memory polymers, the film samples were tested in tension mode at room temperature. The stress–strain curves for the tensile tests of the PSX750/PBA-a blends at 30, 40, and 50 wt% are exhibited in Figure 5. It can be seen that the tensile properties with blends were slightly increased with benzoxazine contents, i.e., the tensile strength values were reported to be 0.23 ± 0.07, 1.02 ± 0.85, and 1.77 ± 0.33 MPa, respectively, while the tensile modulus values were reported at 11.62 ± 67.88, 218.85 ± 111.64, and 439.88 ± 132.59 MPa, respectively. The tensile strength and modulus of pure PBA-a were reported at 35 MPa and 3.3 GPa, respectively [32]. In addition, the strain at break of the blends tended to be increased with the decrease in benzoxazine contents, i.e., ca. 2.6, 4.1, and 5.2% for 50, 40, and 30 wt%, respectively. This could be due to the addition of PSX750, which could lead to more elasticity from greater flexibility of the PSX750 domain in nature. The phenomena indicated that the addition of benzoxazine could provide the strength of the blends. The glass transition temperature values, determined from the peak of the tan delta in the dynamic mechanical analysis technique, are presented in Table 1. Comparatively, the tensile strength values of the blends in this study were in the range of shape memory polymers based on coordinative polysiloxane functionalized with the 2-hydroxy-1-naphthyl imine group, i.e., ca. 0.3–8.0 MPa; however, the strain of the PSX750/PBA-a blend was found to be much lower than that of the elastomers in the report [21,37]. Therefore, our blending system could be proper for some applications that require low strain.

### 3.6. Water Absorption Behaviors

Water absorption or moisture absorption is the capacity of the polymer to absorb water from its environment. It could have an effect on the physical properties and the applications of the materials in some given conditions. The water absorption of the blends after 1-day water immersion is presented in Figure 6. It could be noticed that the addition of benzoxazine could lead to a decrease in water absorption, i.e., 1.9% (for 10 wt%), 1.8% (for 20 wt%), 1.4% (for 30 wt%), 0.7% (for 40 wt%), and 0.4% (for 50 wt%). Polybenzoxazine possesses much lower water absorption than pure PSX750. Although the structure of polybenzoxazine contains polar functional groups, this kind of thermoset shows low water absorption because of the complex hydrogen bond formation between their network structures [38]. Therefore, the water absorption values of pure polybenzoxazine and PSX750 were 0.2 and 2.4%, respectively.

## 4. Conclusions

In this study, polybenzoxazine, a novel type of phenolic thermoset with many beneficial properties, was developed by blending with diamino polysiloxane to reduce the brittleness of the polymers. The shape memory behaviors and physical and thermal properties were determined. It could be found that the homogenous blends with benzoxazine (BA-a) content of 0–50 wt% were observed. The hydrogen bonds between two polymeric components were noticed from the shift of FTIR spectra. The decomposition temperature increased with the addition of PSX750 while the char yield increased with PBA-a. The blends at the content of 10 and 20 wt% could switch the shape at only low shape recovery temperature (T_gL_) in the range of −63 to −70 °C. The shape recovery ratio at low recovery temperature was 100% with the recovery time being 60 and 77 s. The multiple shape recovery behaviors were observed at the content ranging from 30 to 50 wt%. The shape recovery ratios at high recovery temperature were reported at approximately 98–99% with the recovery time in the range from 57 to 65 s. The shape recovery values at a low recovery temperature were approximately 99% with a recovery time in the range from 78 to 360 s. Multiple-shape recovery behaviors with a high recovery ratio in the range of 95–98% could be observed even though four shape-recovery cycles were conducted. This could indicate the reusability of the multiple-shape recovery blends.

## Figures and Tables

**Figure 1 polymers-15-03814-f001:**
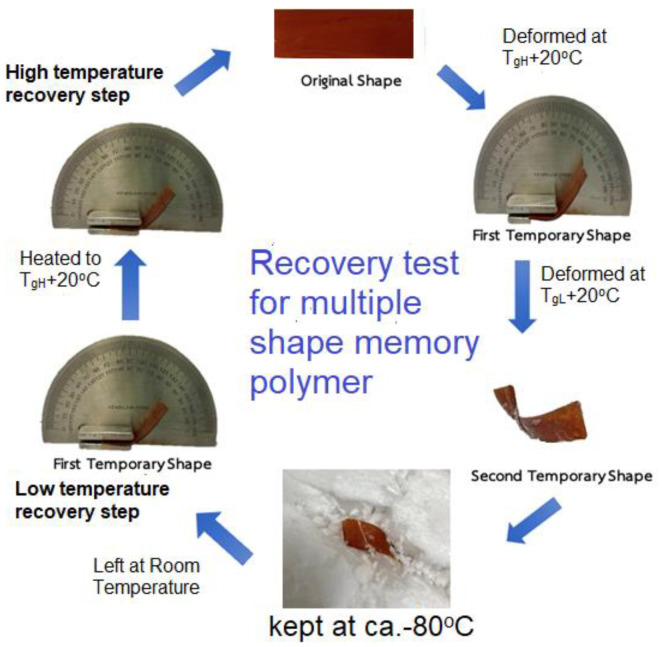
Cycle of recovery test for multiple shape memory polymer.

**Figure 2 polymers-15-03814-f002:**
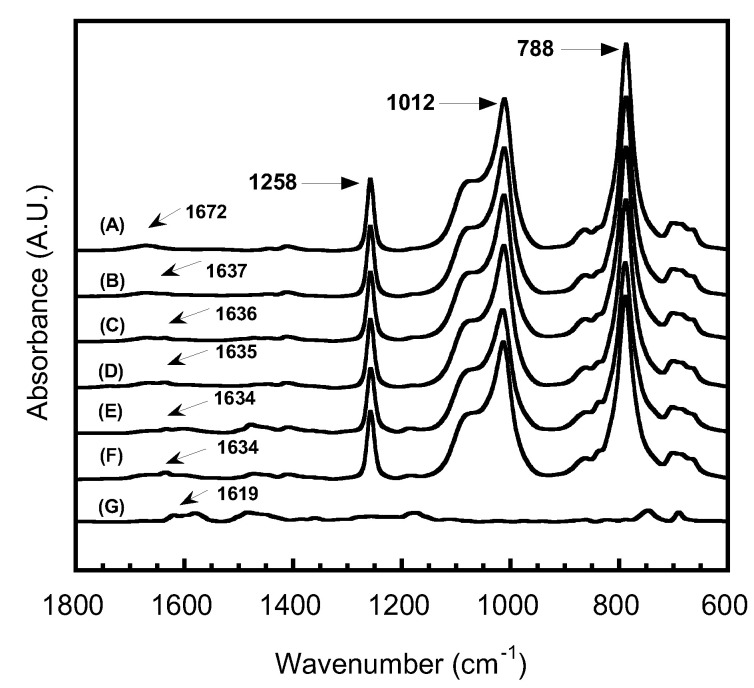
ATR FT-IR spectra of PSX750/PBA-a blends at different benzoxazine contents: (A) Pure PSX750, (B) 10%, (C) 20%, (D) 30%, (E) 40%, (F) 50%, and (G) pure PBA-a.

**Figure 3 polymers-15-03814-f003:**
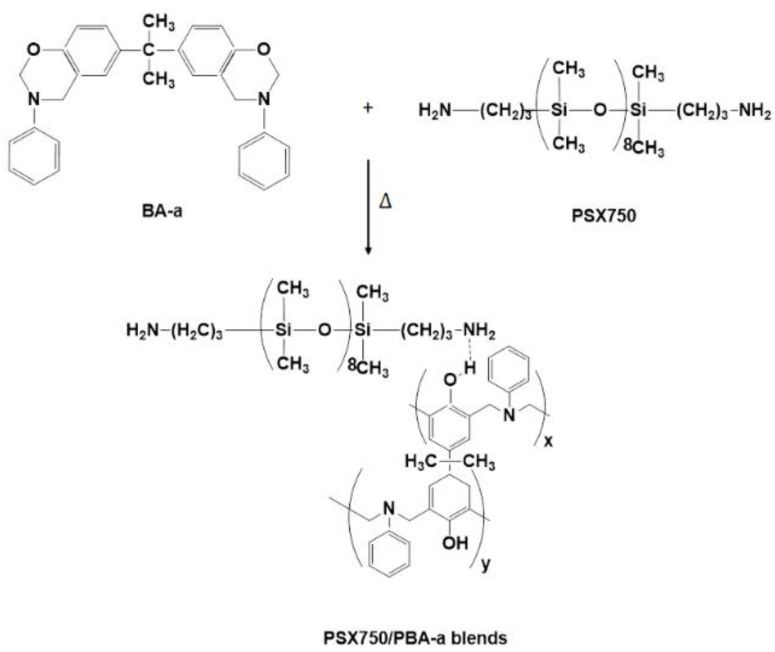
Proposed molecular structure of PSX750/PBA-a blends.

**Figure 4 polymers-15-03814-f004:**
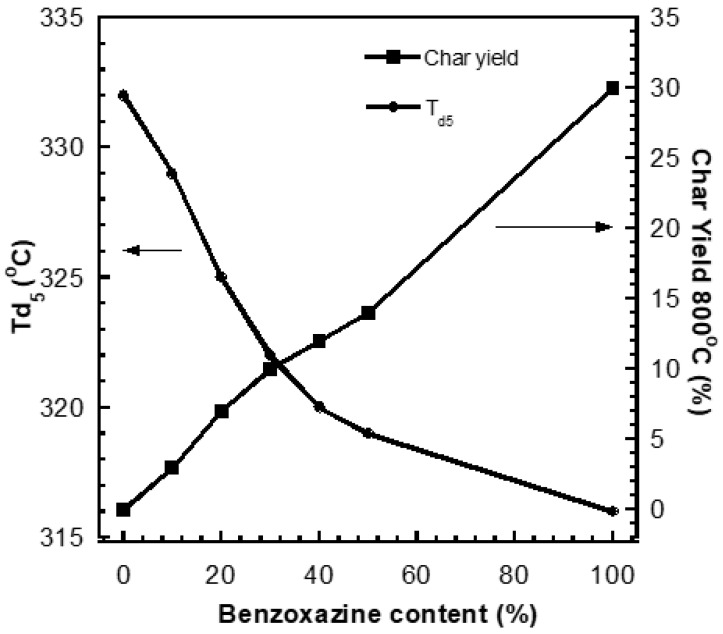
Decomposition temperature at 5% weight loss (T_d5_) and char yield at 800 °C of the PSX750/PBA-a blends at various benzoxazine contents.

**Figure 5 polymers-15-03814-f005:**
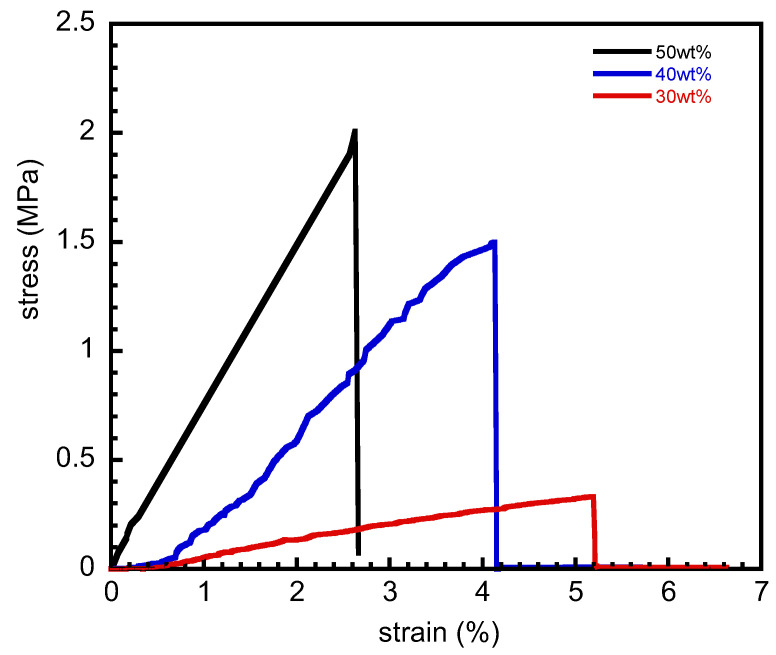
Stress–strain curves of the multiple-shape memory polymers at 30, 40, and 50 wt% benzoxazine contents.

**Figure 6 polymers-15-03814-f006:**
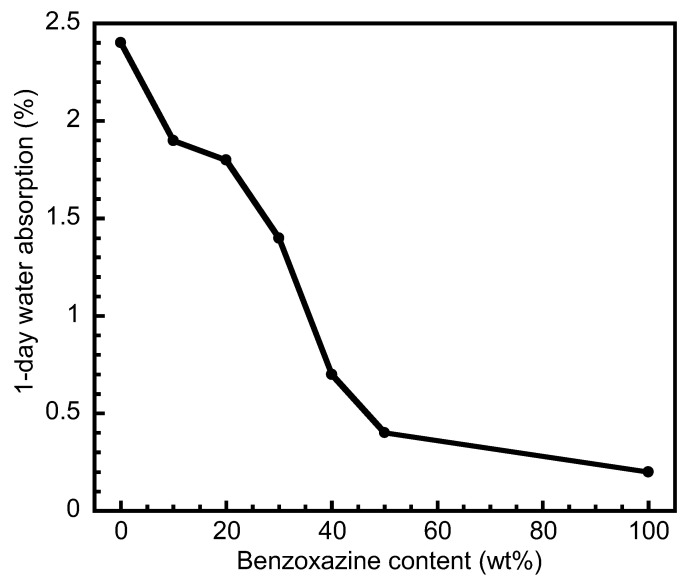
Water absorption of the blends at various benzoxazine contents.

**Table 1 polymers-15-03814-t001:** Glass transition temperatures of the blends at various benzoxazine contents.

BA-a Content (%)	T_gL_ (°C)	T_gH_ (°C)
10	−70	N/A
20	−64	N/A
30	−46	150
40	−23	157
50	−17	165

**Table 2 polymers-15-03814-t002:** Type of shape memory behaviors, shape recovery ratio, shape fixity, and recovery time at high and low recovery temperatures of the blends at various benzoxazine contents.

Benzoxazine Content (%)	Type of Shape Memory Behavior	Shape Recovery Ratio at High Recovery Temperature (%)	Shape Fixity Ratio at High Recovery Temperature (%)	Shape Recovery Ratio at Low Recovery Temperature (%)	Recovery Time at High Recovery Temperature (Seconds)	Recovery Time at Low Recovery Temperature (Seconds)
10	Dual	N/A	N/A	100	N/A	60
20	Dual	N/A	N/A	100	N/A	77
30	Triple	99.4	25.6	99.4	57	78
40	Triple	99.3	40.0	99.0	63	120
50	Triple	97.5	66.7	98.9	65	360

**Table 3 polymers-15-03814-t003:** Recovery ratio for the blends at different benzoxazine contents after various shape-recovery cycles.

Cycle Number	Recovery Ratio at Low Shape Recovery Temperature			Recovery Ratio at High Shape Recovery Temperature		
	Benzoxazine Content (%)			Benzoxazine Content (%)		
	30	40	50	30	40	50
1	99.4	99.0	98.9	99.4	99.3	97.5
2	98.9	99.0	97.8	99.4	98.7	97.7
3	98.3	98.0	95.6	98.8	98.1	97.0
4	97.2	98.0	95.0	96.5	96.3	96.3

## Data Availability

Data can be provided upon request from corresponding author.

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
