# Peer review of "Investigation of Multiple Shape Memory Behaviors, Thermal and Physical Properties of Benzoxazine Blended with Diamino Polysiloxane"

_polymers, 2023, doi:10.3390/polym15183814_

Round 1

Reviewer 1 Report

The authors investigated shape memory behaviors, thermal and physical properties of benzoxazine blended with diamino polysiloxane. Significant results were observed such as the multiple shape recovery behaviors with high recovery ratio in the range of 95-98%. The topic is original, and relevant in the field. 

1. Make the figures of the stress-strain curve by each experiments.

2. Table1, and Table 2 : Is the recovery ratio a result of once, or multiple times?  What happens to the recovery ratio for 5 times or 10 times?

3. Describe temperatures of low recovery temperature and high recovery temperature clearly each.

4. Reference 20 : mistake of surname of one author ; Takeok, S.  --->Takeoka, S.

Author Response

  1. Make the figures of the stress-strain curve by each experiment.

Reply to the reviewer: Figure of the stress-strain curves for multiple shape memory polymer were exhibited as shown in Figure 5 as the suggestion.

  1. Table 1 and Table 2: the recovery ratio of a result of once or multiple times? What happens to the recovery ratio for 5 times or 10 times:

Reply to the reviewer: the explanation of the recovery ratio in Table 1 and Table 2 has been included in the revised manuscript. For the recovery ratio for 5 times or 10 times could not be presented because the materials did not present shape memory behaviors in case of 10 recovery cycles. This was mentioned in the revised manuscript.

  1. Describe temperatures of low recovery temperature and high recovery temperature clearly each.

Reply to the reviewer: The explanations has been given in the revised version as suggested.

  1. Reference 20: Mistake of surname of one author: “Takeok, S.” should be “Takeoka S.”.

Reply to the reviewer: The surname of one author in Reference 20 has been revised as suggested.

Reviewer 2 Report

This research studied the shape memoty related properties of the mixtures of benzoxazine and diamino polysiloxane. The authors blended these two with different ratios and measured the FT-IR, thermal stability as well as the glass transition temperatures to investigate the structures of the blended materials. They also studied the shape memory behaviors of those mixtures.

I think this study is pretty straightforward. All the experimental results showed the consistent behaviors as well. To better improve the article, my suggestions are:

1. If possible, present some literature studies on similar materials and compare their properties with the BA-a/PSX750 mixtures so that readers can understand more about the creativity and significance of the study.

2. Try to pay more attention to the format. Line 39, some words are in italic. In figure 1, the font size of the notations are different.  Figure 4 and figure 5, it's better to add the legends for those lines.

Author Response

Thank you very much for taking the time to review this manuscript. Please find the detailed responses below and the corresponding revisions highlighted in the re-submitted files.

  1. If possible present some literature studies on similar materials and compare their properties with the BA-a:PSX750 mixtures so that readers can understand more about the creativity and significance of the study

Reply to the reviewer: More literature studies and comparison with other similar systems have been mentioned in the manuscript as suggested.

  1. Try to pay more attention to the format. Figure 4 and Figure 5; It is better to add the legends for those lines.

Reply to the reviewer:  The format has been checked; the legends for Figures 4 and 5 have been added as suggested.

Reviewer 3 Report

The paper reports the preparation and characterizations of Benzoxazine and polysiloxane blends with Shape Memory Behaviors. It can be published after a few minor revisions are made:

1)    Page 4, line 153-155, the blending systems refer to the blend before curing? How much THF do the blends have? How was the phase separation been determined? Any morphology studies before and after curing?

2)    Since the Benzoxazine is the minor phase, polysiloxane is the matrix, which component forms the continuous phase ?

3)    What are the atmosphere and flow rate used for TGA?

4)    Could the partial miscibility between PSX750/PBA-a be explained further with more references? PDMS is well known to have high Flory Huggins parameter with other polymers.

5)    Section 3.5: Please consider to add a table or figure to summarize the mechanical properties.

6)    Please consider to cite “ACS Appl. Polym. Mater. 2020, 2, 12, 5835–5844”.

7)    Fix minor grammatical errors. E.g. “page 7 line 219 PSX750/PBA-a blends are partially miscibility was shown”.

Author Response

Thank you very much for taking the time to review this manuscript. Please find the detailed responses below and the corresponding revisions highlighted in the re-submitted files.

1) Page 4, line 153-155, the blending systems refer to the blend before curing? How much THF do the blends have? How was the phase separation been determined? Any morphology studies before and after curing?

Reply to the reviewer:   The systems were blended before thermal curing. The solution of PSX750 were prepared from mixing 2 g of PSX750 in 10 mL of THF. The sea-island appearances were observed to indicate phase separation. Any other morphology studies would be further studied in the future.

2) Since the Benzoxazine is the minor phase, polysiloxane is the matrix, which component forms the continuous phase?

Reply to the reviewer:   This blending system was supposed to be partially miscible not the compositing system.

3) What are the atmosphere and flow rate used for TGA?

Reply to the reviewer:   The details about TGA analysis was provided as suggested.

4) Could the partial miscibility between PSX750/PBA-a be explained further with more references? PDMS is well known to have high Flory Huggins parameter with other polymers.

Reply to the reviewer:  The system between silicon-containing polyimide and polybenzoxazine in our previous work was an example of partially miscibility.

5) Section 3.5: Please consider to add a table or figure to summarize the mechanical properties.

Reply to the reviewer: Figure 5 about the stress-strain curves is added as the suggestion of reviewer.   

6)    Please consider to cite “ACS Appl. Polym. Mater. 2020, 2, 12, 5835–5844”.

Reply to the reviewer: The mentioned reference was cited as suggested.

7) Fix minor grammatical errors. E.g. “page 7 line 219 PSX750/PBA-a blends are partially miscibility was shown”.

Reply to the reviewer: The sentence was revised as suggested.
